# Exploring the Molecular Mechanisms of Macrolide Resistance in Laboratory Mutant *Helicobacter pylori*

**DOI:** 10.3390/antibiotics13050396

**Published:** 2024-04-26

**Authors:** Meltem Ayaş, Sinem Oktem-Okullu, Orhan Özcan, Tanıl Kocagöz, Yeşim Gürol

**Affiliations:** 1Department of Medical Laboratory Techniques, Vocational School of Health Services, Acibadem Mehmet Ali Aydinlar University, 34752 Istanbul, Turkey; 2Department of Medical Biotechnology, Institute of Health Sciences, Acibadem Mehmet Ali Aydinlar University, 34752 Istanbul, Turkey; tanil.kocagoz@acibadem.edu.tr; 3Department of Medical Microbiology, Faculty of Medicine, Acibadem Mehmet Ali Aydinlar University, 34752 Istanbul, Turkey; sinem.oktem@acibadem.edu.tr (S.O.-O.); yesim.gurol@acibadem.edu.tr (Y.G.); 4TrioScience Biotechnology, 34000 Istanbul, Turkey; orhn.ozcn@hotmail.com

**Keywords:** *H. pylori*, macrolide resistance, clarithromycin, next-generation sequencing

## Abstract

Resistance to clarithromycin, a macrolide antibiotic used in the first-line treatment of *Helicobacter pylori* infection, is the most important cause of treatment failure. Although most cases of clarithromycin resistance in *H. pylori* are associated with point mutations in 23S ribosomal RNA (rRNA), the relationships of other mutations with resistance remain unclear. We examined possible new macrolide resistance mechanisms in resistant strains using next-generation sequencing. Two resistant strains were obtained from clarithromycin-susceptible *H. pylori* following exposure to low clarithromycin concentrations using the agar dilution method. Sanger sequencing and whole-genome sequencing were performed to detect resistance-related mutations. Both strains carried the A2142G mutation in 23S rRNA. Candidate mutations (T1495A, T1494A, T1490A, T1476A, and G1472T) for clarithromycin resistance were detected in the Mutant-1 strain. Furthermore, a novel mutation in the gene encoding for the sulfite exporter TauE/SafE family protein was considered to be linked to clarithromycin resistance or cross-resistance, being identified as a target for further investigations. In the Mutant-2 strain, a novel mutation in the gene that encodes DUF874 family protein that can be considered as relevant with antibiotic resistance was detected. These mutations were revealed in the *H. pylori* genome for the first time, emphasizing their potential as targets for advanced studies.

## 1. Introduction

Approximately 50% of the world’s population is colonized by *Helicobacter pylori*, a Gram-negative, spiral-shaped bacterium that can survive in the acidic environment of the stomach. The International Agency for Research on Cancer has identified *H. pylori* as a Group 1 carcinogen that is capable of causing gastric cancer [1,2]. Furthermore, it has been reported that eradicating *H. pylori* is an effective strategy for treating peptic ulcers and gastric mucosa-associated lymphoid tissue lymphoma, as well as for preventing gastric cancer [3].

Several factors, such as socioeconomic status and environmental conditions, have been linked to the risk of *H. pylori* infection at early ages [4,5,6,7]. Therefore, although the incidence of *H. pylori* infection is decreasing in industrialized countries, it remains stable in developing countries [8].

*H. pylori* is typically acquired early in childhood. Although the prevalence of *H. pylori* infection is higher in the first five years of life in developing countries, there have been reports of decreased infection rates due to improved hygiene and sanitation [9,10]. Conversely, the risk of infection is low in adulthood [8,11,12]. Current estimates suggest that approximately 4.4 billion individuals worldwide are infected with *H. pylori*. In European countries, the prevalence of *H. pylori* has decreased since 2000; however, it remains unchanged in Asia. Reports indicate that Africa has the highest prevalence (70.1%), whereas Switzerland has the lowest (18.9%). Among Southern Asian countries, Pakistan and India exhibit the highest prevalence of *H. pylori* (81% and 63.5%, respectively). In Western Asia, Turkey has the highest prevalence (77.2%) [13].

The presence of *H. pylori* in saliva, vomit, and stool samples has been observed, but there is no evidence supporting these bodily fluids as the predominant mode of transmission. Person-to-person transmission through gastro–oral, oral–oral, or fecal–oral exposure appears likely. Because *H. pylori* has a limited host range, new infections arise via person-to-person transmission or environmental contamination. It remains unclear which of the aforementioned transmission routes is the most frequent [14,15,16,17]. Hereditary susceptibility to *H. pylori* infection has also been considered, but it has not been proven [18].

According to current international guidelines on *H. pylori* treatment, first-line therapy includes proton pump inhibitors (PPIs), clarithromycin, and amoxicillin/metronidazole. Although this therapy is considered standard, treatment failure due to increased antibiotic resistance, particularly to clarithromycin, has been observed. Consequently, guidelines do not recommend the standard therapy in regions where clarithromycin resistance rates exceed 15% [19,20,21,22,23,24]. With increasing resistance rates, recommending standard triple therapies as empiric therapy may no longer be feasible. Given the high level of resistance to clarithromycin and metronidazole, along with varied resistance patterns across populations, it is crucial to design standard triple therapies to address local resistance patterns. Wherever possible, treatment decisions should be based on strain culture susceptibility information. Various alternative treatment strategies are currently employed in clinical practice to manage antibiotic-resistant *H. pylori* strains. Consequently, novel effective treatments with greater efficacy, including probiotics, have been developed and utilized to enhance eradication regimens and mitigate antibiotic-associated side effects [25].

Clarithromycin, a macrolide antibiotic, is the most potent bacteriostatic antibiotic used to treat *H. pylori* infection. Because many international guidelines recommend clarithromycin-containing bismuth quadruple therapies for *H. pylori* treatment, clarithromycin remains important in treating *H. pylori* infections [19,20,21]. In recent years, clarithromycin resistance has become more prevalent worldwide. The resistance rate in Europe increased from 9% to 17.6% between 1998 and 2008, while that in Japan increased from 7% to 27.7% between 2000 and 2006 [12]. Clarithromycin resistance rates in Turkey have been estimated to have reached approximately 40% [26]. The World Health Organization recently published a list of bacteria requiring new antibiotics urgently, and clarithromycin-resistant *H. pylori* was included in the high-priority list [27].

Inhibiting protein synthesis is the main mechanism of action of clarithromycin, which binds to receptors located in the 23S ribosomal subunit of the 50S ribosome. Point mutations in the peptidyl transferase encoded in domain V of 23S ribosomal RNA (rRNA) are responsible for clarithromycin resistance [28]. Several point mutations in the 23S rRNA gene have been reported to decrease the drug’s affinity for ribosomes, causing the bacteria to become resistant. More than 90% of clarithromycin-resistant strains harbor three point mutations in the peptidyl transferase region of domain V of 23S rRNA. These mutations include a substitution from adenine to guanine at position 2143 (A2143G) and one from adenine to guanine or cytosine at position 2142 (A2142G or A2142C) [29,30,31,32,33]. Several other point mutations in the region contribute to clarithromycin resistance; however, their significance remains unclear [31,32,33]. Another mechanism of clarithromycin resistance involves five efflux pump systems. Current findings suggest that four groups of genes (HP0605–HP0607, HP0971–HP0969, HP1327–HP1329, and HP1489–HP1487) can function as efflux pumps in *H. pylori* [34,35,36]. Despite the major molecular mechanisms of clarithromycin resistance demonstrated in previous studies, it remains unclear whether other gene mutations outside 23S rRNA are associated with resistance in *H. pylori*. The presented study aimed to identify new clarithromycin resistance mechanisms in *H. pylori*. Because there is a need for more information about the clarithromycin resistance mechanism in *H. pylori*, this study provides a key contribution to the literature. The tests currently used in laboratories are designed to detect clarithromycin resistance in *H. pylori* by identifying only the most common mutations (A2143G, A2142G, and A2142C) associated with resistance. Therefore, we predict that the candidate mutations identified in the study could also contribute to the development of molecular diagnostic tests by highlighting new resistance mechanisms in the target regions.

## 2. Results

### 2.1. Clarithromycin Minimum Inhibitory Concentration (MIC) Value for Helicobacter pylori G27

The agar dilution test results were evaluated visually. Bacterial growth was observed at clarithromycin concentrations of 0.015 and 0.03 µg/mL and on the growth control medium. However, no growth was noted at clarithromycin concentrations of ≥0.06 µg/mL. Thus, the MIC of clarithromycin was determined to be 0.06 μg/mL (Figure 1).

### 2.2. Establishment of Resistant Strains

The reference strain *H. pylori* G27, which is susceptible to clarithromycin, was exposed to low concentrations of clarithromycin, and two resistant mutant strains were obtained. Accordingly, Mutant-1 and Mutant-2 exhibited clarithromycin MICs of 128 and >256 μg/mL, respectively (Figure 1).

### 2.3. Sanger Sequencing

Polymerase chain reaction (PCR) results associated with the amplicons are presented in Figure 2. The A2142G mutation was detected in both the Mutant-1 and Mutant-2 strains (Figure 3).

### 2.4. Next-Generation Sequencing (NGS) for Detecting Mutations in the Resistant Strains

The DNAs of *H. pylori* G27 and the mutant strains were sequenced. The resulting sequences were compared between G27 and the mutants. In the Mutant-1 strain, shifts from thymine to adenine at positions 1476, 1490, 1494, and 1495, as well as a shift from guanine to thymine at position 1472 were observed in the 23S rRNA region. Additionally, the A2142G mutation was detected in Mutant-1. Furthermore, a single-nucleotide mutation in a gene encoding the sulfite exporter TauE/SafE family protein was observed. Because of the leucine-to-proline mutation, structural changes occurred in the proteins. In Mutant-2, the A2142G was detected. In addition, mutations were also detected in genes that encodes the DUF874 Family protein in Mutant 2. As a result of these mutations, some aminoacidic changes were observed (Table 1). Bioinformatic analysis data are presented as the Appendix A (Appendix A, Appendix A, Appendix A).

## 3. Discussion

The discovery of *H. pylori* infection and its role in various diseases, including gastric cancer, has completely changed the treatment landscape for infected patients. Currently, antibiotics represent the only available effective therapy for *H. pylori* infection. However, few antibiotics can act against this bacterium, which resides in acidic stomach conditions. The standard triple therapy, including a PPI, clarithromycin, and either amoxicillin or metronidazole, has been used as the first-line treatment for eradicating *H. pylori* in regions where clarithromycin resistance rates are lower than 15%. The success of the treatment relies on factors such as smoking, treatment adherence, and antibiotic resistance. Resistance to clarithromycin has been reported to be the most crucial factor that leads to treatment failure [37,38].

Clarithromycin is a bacteriostatic antibiotic that reversibly binds to the peptidyl transferase loop of domain V of 23S rRNA. In most Gram-negative bacteria, the mechanism of resistance to clarithromycin or other macrolides involves modification of the target region (nucleotide mutations and methylation of 23S rRNA) or overexpression of efflux pumps [39].

Macrolides exert bacteriostatic effects by inhibiting protein synthesis via reversible binding to the peptidyl transferase loop of domain V of 23S rRNA in the 50S subunit [40]. Clarithromycin is the most preferred macrolide for *H. pylori* treatment because of its pharmacokinetic advantages, including stability under acidic conditions and better absorption in the gastric mucus layer [41]. The World Health Organization has reported clarithromycin-resistant *H. pylori* as a high-priority pathogen that requires attention in treatment [42]. Generally, Gram-negative bacteria can develop resistance to macrolides through three mechanisms: methylation of the target cells or mutations of critical genes, efflux mechanisms, and enzymatic deactivation of the drug. In *H. pylori*, specific mutations in the 23S rRNA region have been linked to clarithromycin resistance. Three primary 23S rRNA mutations have been reported, namely A2142G, A2143G, and A2142C, with prevalence rates of 11.7%, 69.8%, and 2.6%, respectively. Numerous other point mutations have also been identified, such as A2115G, T2117C, G2141A, T2182C, G2224A, C2245T, T2289C, C2611A, and T2717C. Despite their low frequencies, the clinical relevance of these mutations is unproven. However, correlations of T2182C, C2611A, and T2717C with low resistance levels have been reported [30,43]. Furthermore, recent studies have found that additional resistance mutations affecting the ribosomal protein L22p and the translation initiation factor IF2 are observed in experimentally induced clarithromycin resistance phenotypes. However, these mutations have not been reported in clinical isolates [44,45].

In the present study, two clarithromycin-resistant strains were successfully produced through exposure to low concentrations of clarithromycin, and novel candidate genes related to clarithromycin resistance were identified using NGS. The A2142G mutation, which is frequently reported to be associated with high clarithromycin MICs in the literature [37,44], was detected in the 23S rRNA region in both mutant strains.

In the Mutant-1 strain, six single-nucleotide mutations at different positions in 23S rRNA were detected via NGS, including A2142G. Sanger sequencing of the clarithromycin target region was performed to confirm the known mutation in mutant strains. The other mutations (T1476A, T1490A, T1494A, T1495A, and G1472T) were detected outside the peptidyl transferase region via whole-genome sequencing (WGS). To the best of our knowledge, these mutations have not been reported previously in the literature. Ultimately, all five candidate mutations should also be confirmed by Sanger sequencing. Furthermore, the relationships of these mutations with clinical clarithromycin resistance should be studied using clinical strains or samples in future studies. Clarithromycin-resistant strains with no 23S rRNA mutation have also been reported, indicating that unknown genes outside 23S rRNA are associated with clarithromycin resistance in *H. pylori*. Binh et al. discovered two novel gene mutations (infB and rpl22) related to clarithromycin resistance in *H. pylori* that exert synergistic effects with 23S rRNA mutations, resulting in higher MICs [37]. However, such mutations were not detected in the present study. A gene mutation as a novel candidate mechanism was observed in one clarithromycin-resistant mutant strain in the present study. This gene encodes the sulfite exporter TauE/SafE family protein. The mutation resulted in an altered amino acid sequence of the protein, with leucine being changed to proline. Weinitschke et al. reported that TauE/SafE proteins act as sulfite/organosulfonate exporters in the metabolism of C2 sulfonates [45]. Furthermore, Rakitin et al. compared the genomes of clinical and permafrost strains of *Acinetobacter lwoffii* and detected two copies of the gene encoding the TauE/SafE family sulfite exporter in environmental isolates whereas clinical strains lacked this gene duplication [46]. Considering the widespread occurrence of organosulfonates in nature, the researchers likely hypothesized that permafrost strains may utilize organosulfonates [46]. To the best of our knowledge, the relationship between sulfite exporter TauE/SafE family proteins and antibiotic resistance has not been reported for any bacterium. Based on this knowledge, if clarithromycin is found in its anionic form, it can be considered that the sulfite exporter TauE/SafE family protein might function as a drug efflux pump due to its role as an anion transporter. Consequently, mutations in this protein could lead to drug resistance. Conversely, previous studies have identified organic anion transporter family members 1B1 and 1B3 as targets of clarithromycin in mammalian cells. It has been reported that clarithromycin inhibits these proteins, leading to interactions with substrate drugs [47]. Hence, it can also be predicted that inhibition of the sulfite exporter TauE/SafE family protein by a high concentration of clarithromycin may result in cross-resistance to substrate anionic antibiotics such as fluoroquinolones, tetracyclines, ampicillin, and rifampicin, which are commonly used to treat *H. pylori* infection. Future studies assessing the antibiotic susceptibility of *H. pylori* G27 and its mutants to anionic antibiotics are needed to confirm this hypothesis.

In the Mutant-2 strain, single point mutations were identified in two consecutive genes in different positions that encode the DUF874 family protein in the genome. These mutations resulted in changes in amino acid structures. It is worth noting that these mutations, particularly due to amino acid alterations, can induce changes in the three-dimensional structure of the protein, signifying their significance (Appendix A, Appendix A).While studies in the literature have indicated an unclear understanding of the function of the DUF874 family protein in *H. pylori*, recent research suggests that this protein may play a crucial role in *H. pylori*’s adaptation to stress conditions within host cells by synthesizing DNA-binding proteins. In a study using mouse models, these proteins have been associated with the survival of *H. pylori* bacteria in the host mouse [48]. To the best of our knowledge, no study demonstrated that clarithromycin resistance was associated with DUF874 family protein mutations in *H. pylori*. We anticipate that our findings will pave the way for further studies via verification in other studies.

Additionally, during the bioinformatics analysis of the sequencing results, some rRNA strand breaks were detected, and, surprisingly, these breaks occurred in the same regions. They were predicted as hypothetically methylated sites and likely cause resistance to antibiotics by decreasing clarithromycin affinity and changing the ribosomal structure (Appendix A, Appendix A). In the literature, resistance to clarithromycin and other macrolides in Gram-positive and Gram-negative bacteria has been linked to post-transcriptional methylation of 23S rRNA [32,39]. However, to the best of our knowledge, no study has revealed the relationship between RNA methylation and clarithromycin resistance in *H. pylori*. Further research is required to investigate whether the 23S rRNA region in *H. pylori* exhibits differential methylation patterns to confirm this claim.

In conclusion, this study identified novel clarithromycin resistance mechanisms in *H. pylori*. The newly discovered mutations in the 23S rRNA region could serve as targets for molecular diagnosis of clarithromycin resistance pending validation studies. Additionally, hypothetically methylated regions in 23S rRNA were demonstrated in the mutant genome for the first time, underscoring the need for further studies to validate these findings. Identifying these alternative genetic mechanisms of reduced susceptibility could facilitate the development of accurate and reliable sequence-based diagnostics for predicting clarithromycin susceptibility and aid in the design of novel therapeutics.

## 4. Materials and Methods

### 4.1. Bacteria

*H. pylori* G27, which has been extensively used in research because of its relatively straightforward genetics and reliable growth, was used for this study. The bacterium was stored at −80 °C in Brucella broth containing 25% (*v/v*) glycerol.

### 4.2. Subculture of H. pylori

First, *H. pylori* was subcultured in Colombia blood agar (Oxoid, Hampshire, United Kingdom) to recover the bacteria from the frozen environment. The agar included 5% defibrinated horse blood and antibiotic cocktails. The agar was dissolved in distilled water according to the manufacturer’s instructions and then autoclaved at 121 °C for 15 min in the liquid sterilization mode. Thereafter, the agar was kept in a water bath at 50 °C for 1 h to achieve a suitable temperature for adding antibiotics and blood. Antibiotic cocktails and freshly prepared β-cyclodextrin were added to the medium to prevent contamination and contribute to bacterial growth. Finally, horse blood was added, and 20 mL of agar was poured into each Petri dish. *H. pylori* was streaked onto the prepared agar plates and incubated at 37 °C for 3 days under microaerophilic conditions (5% O_2_, 10% CO_2_, and 85% N_2_) provided by an automated closed system. After incubation, Gram staining was performed to eliminate the possibility of contamination. All processes using *H. pylori* were conducted in a biosafety level II cabinet in the microbiology laboratory.

### 4.3. Agar Dilution Susceptibility Tests

Agar dilution susceptibility tests were performed according to the Clinical and Laboratory Standards Institute (CLSI) guidelines [49]. According to these guidelines, the resistance breakpoint for clarithromycin is >0.5 µg/mL [49]. First, 2-fold serial dilutions of clarithromycin were prepared in 2 mL of broth. Then, 1 mL of sterile defibrinated sheep blood was added to each tube (to obtain 5% horse blood in 20 mL). Finally, 17 mL of Mueller–Hinton agar (Becton Dickinson and Company, Franklin Lake, NJ, USA) was added. Therefore, the initial concentration of clarithromycin was diluted 10-fold, with final concentrations of 0.015–256 μg/mL achieved on Petri dishes.

### 4.4. Inoculation of Bacteria and Interpretations of Results

A 2.0 McFarland bacterial suspension was prepared in sterile saline, and 3 mL of the suspension was inoculated onto each plate at different antibiotic concentrations. After 3 days of incubation under microaerophilic conditions, the lowest concentration of the related antibiotic that prevented visible growth of the bacterium was defined as the MIC. According to the European Committee on Antimicrobial Susceptibility Testing and CLSI guidelines, the resistance breakpoint for clarithromycin is >0.5 mg/mL [49,50]. Agar dilution susceptibility tests were performed thrice.

### 4.5. In Vitro Selection of Clarithromycin-Resistant Strains

*H. pylori* G27, which is susceptible to clarithromycin, was used as the parent strain. Clarithromycin-resistant strains were generated from this strain via exposure to low concentrations of clarithromycin in vitro using the agar dilution method. A single colony of G27 was inoculated onto Mueller–Hinton agar supplemented with 5% defibrinated sheep blood without antibiotics. The plate was incubated at 37 °C under microaerophilic conditions (10% O_2_, 5% CO_2_, and 85% N_2_) for 72 h. Colonies on agar plates were harvested, placed in sterile saline, and exposed to serially doubling concentrations of clarithromycin (0.03–256 µg/mL) using the agar dilution method. Then, colonies from the highest concentrations under which bacteria could grow were obtained and repeatedly transferred to the same clarithromycin concentration five times before being exposed to a higher concentration. Individual colonies on the plates containing a clarithromycin concentration greater than the MIC were isolated [37]. In other words, clarithromycin MIC determination and mutant selection were performed simultaneously.

### 4.6. DNA Isolation from H. pylori G27 and Mutant Strains

Using the High Pure PCR template preparation kit (Roche, Basel, Switzerland), the genomic DNA of *H. pylori* was isolated according to the manufacturer’s instructions. DNA quantification and quality control were completed immediately after isolation using a NanoDrop One/OneC Spectrophotometer (Thermo Fisher Scientific, Waltham, MA, USA). The DNA samples’ ratio of absorbance at 260/280 nm (OD 1.8–2.0) and 260/230 nm (OD 2–2.2) was considered for DNA sequencing analysis.

### 4.7. Sanger Sequencing

Specific primers covering all putative mutations associated with clarithromycin resistance were used to target regions. The primers and PCR conditions are described in Table 2 [51]. PCR amplicons were sent to Eurofins Scientific (Luxembourg, Germany) for Sanger sequencing. The DNA sequences were edited using BioEdit version 7.2.5 and aligned using BLAST [52,53].

### 4.8. WGS and Bioinformatics Analysis

WGS was performed using Illumina NGS Platforms (Illumina, San Diego, CA, USA) by Eurofins Scientific (Luxembourg). Candidate mutations associated with clarithromycin resistance were obtained by comparing the reconstructed genomes of the mutant strains with that of the *H. pylori* G27 strain. Bioinformatics analysis was performed using the tools listed in Table 3.

## Figures and Tables

**Figure 1 antibiotics-13-00396-f001:**
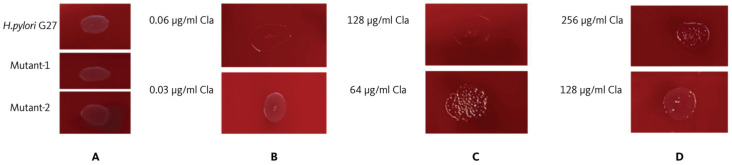
Agar dilution assay for assessing the clarithromycin susceptibility of the strains. (**A**): Growth control, (**B**): Clarithromycin MIC value of the *H. pylori* G27 strain, (**C**): Clarithromycin MIC value of the Mutant-1 strain, (**D**): Clarithromycin MIC value of the Mutant-2 strain. Cla: Clarithromycin.

**Figure 2 antibiotics-13-00396-f002:**
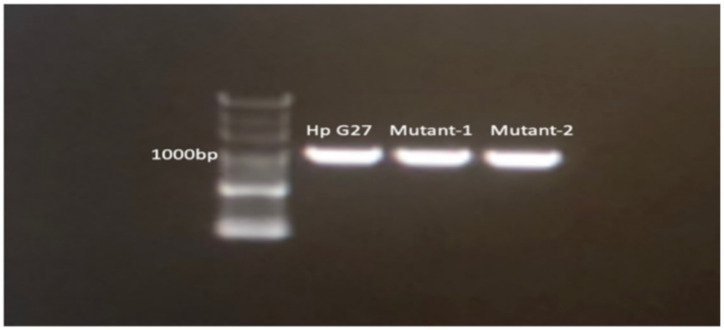
PCR amplification of the *H. pylori* G27, Mutant-1, and Mutant-2 strains and 1% agarose gel electrophoresis.

**Figure 3 antibiotics-13-00396-f003:**
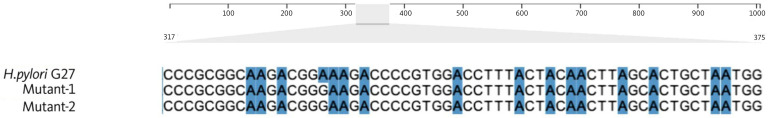
Multiple-sequence alignment of 23S rRNA gene sequences of the *H. pylori* G27, Mutant-1, and Mutant-2 strains compared to a reference gene.

**Table 1 antibiotics-13-00396-t001:** Candidate mutations associated with clarithromycin resistance in mutant strains detected via whole-genome sequencing.

Gene	Position of Mutations	Mutation Type	*H. pylori* G27	Mutation	Mutant-1	Mutant-2	AA Change
23SrRNA	2142	SNP	A	G	+	+	-
23SrRNA	1490	SNP	T	A	+	−	No
23SrRNA	1494	SNP	T	A	+	−	No
23SrRNA	1495	SNP	T	A	+	−	No
23SrRNA	1476	SNP	T	A	+	−	No
23SrRNA	1472	SNP	G	T	+	−	No
Sulfite exporter TauE/SafE family protein	35	SNP	A	G	+	−	L > P
DUF874 Family proteinGene-1	589	SNP	T	C	−	+	H > R
DUF874 Family proteinGene-1	587	SNP	G	A	−	+	No
DUF874 Family proteinGene-1	572	SNP	G	T	−	+	No
DUF874 Family proteinGene-1	553	SNP	G	A	−	+	T* > I
DUF874 Family proteinGene-1	549	SNP	T	G	−	+	T* > A
DUF874 Family proteinGene-2	941	SNP	T	C	−	+	No
DUF874 Family proteinGene-2	890	SNP	C	T	−	+	No
DUF874 Family proteinGene-2	703	SNP	G	A	−	+	A** > V
DUF874 Family proteinGene-2	701	SNP	G	A	−	+	No

SNP: single nucleotide polymorphism, A: Adenine, T: Thymine, G: Guanine, C: Cytosine, AA: Amino acids, L: Leucine, P: Proline, H: Histidine, R: Arginine, T*: Threonine, I: Isoleucine, A**: Alanine, V: Valine. +: a nucleotide mutation occurred, −: no nucleotide mutation occurred. No: No aminoacid change occurred.

**Table 2 antibiotics-13-00396-t002:** Primers and PCR conditions used for obtaining PCR amplicons.

Primers	PCR Conditions
23S-F 5′-AGCACCGTAAGTTCGCGATAAG-3′	Initial denaturation: 94 °C for 4 min 30 cycles at 94 °C for 1 min 56 °C for 45 s 72 °C for 10 min
23S-R 5′-CTTTCAGCAGTTATCACATCC-3′	

**Table 3 antibiotics-13-00396-t003:** Bioinformatics tools used for whole-genome sequencing.

Tools	Intended Purpose	References
BBtools v39.01—BBDuk	Quality trimming and filtering	[54]
Spades v3.15.4	Genome assembly	[55]
Dfast v1.6.0	Annotation of the assembled contigs	[56]
BBtools v39.01—BBmap	Mapping of *H. pylori* reads to the reference G27 genome	[54]
BBtools v39.01—Call Variants	Variant calling from sam files with callvariants.sh.	[57]
EMBOSS Transeq v6.6.0.0	Conversion of SNP-bearing contigs and regions for phenotypic change	[58]

## Data Availability

The data presented in this article are included as Appendix A.

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
