# Peer review of "Exploring the Molecular Mechanisms of Macrolide Resistance in Laboratory Mutant Helicobacter pylori"

_antibiotics, 2024, doi:10.3390/antibiotics13050396_

Round 1
Reviewer 1 Report
Comments and Suggestions for Authors
Manuscript by AYAÅž et al. Studies of the molecular mechanisms underlying macrolide resistance in laboratory mutant Helicobacter pylori are described. Overall, the manuscript requires extensive editing, and some of the required changes are as follows:
Minor:
Lines 24-28: unclear! Rephrase it.
Line 33: Remove H. pylori.
Lines 35, 44-45, 62-62, and 63-64: Please add a reference.
Line 143: Genomic DNA?
Major:
Figures 1, 2, and 4: Images are blurry and low resolution. Please add more information. Figures 1, 2 and 3 are supplementary data. I suggest creating a diagram for sections 2.1 and 2.2
Authors state: “The genome sequence of the resulting mutant was compared with the H. pylori G27 genome” (line 143). Where are the results?
Line 337: Section 4.4. Whole-genome sequencing and bioinformatics analysis. Where is bioinformatics analysis? Where is the additional information file?
The author should attach all raw data as supplementary data to support his/her results.
All References need more work – there are some odd capitals/lack of capitals.
Comments on the Quality of English LanguageEnglish grammar requires improvement throughout the manuscript.
Author Response
Dear Sir/ Madam
First of all, thank you for your valuable insights and contributions. Following your suggestions, the English revisions of our article have been reviewed and restructured by native English-speaking professionals. Please find the detailed responses at below.
Minor:
Lines 24-28: unclear! Rephrase it.
The paragraph (Lines 24-28) revised as below:
“Furthermore, a novel mutation in the gene that encodes the Sulfite exporter TauE/SafE family protein was regarded as linked to clarithromycin resistance or cross-resistance and identified as a target for further investigations. Additionally, hypothetical methylated regions in the 23S rRNA were revealed in the mutant genome for the first time, emphasizing their potential as targets for advanced studies.”
Line 33: Remove H. pylori.
Indicated word (H. pylori) was removed from the sentence.
Lines 35, 44-45, 62-62, and 63-64: Please add a reference.
For line 35: ref 2 was added.
For line44-45: ref 9-10were added.
For line 62, 63, 64: ref 22,23,24 were added.
Line 143: Genomic DNA?
Since genomic DNA constitutes the total genetic information of the bacteria, we used this term. However, for clear understanding the sentence has been revised as "The DNAs of H. pylori G27 and the mutant strains was sequenced. The resulting sequences were compared between G27 and the mutants".
Major:
Figures 1, 2, and 4: Images are blurry and low resolution. Please add more information. Figures 1, 2 and 3 are supplementary data. I suggest creating a diagram for sections 2.1 and 2.2.
Regarding the reviewer comment, section 1 and Figure 1 remove the manuscript. The figure 2 and 4 has been revised again to be explanatory and of high resolution.
Authors state: “The genome sequence of the resulting mutant was compared with the H. pylori G27 genome” (line 143). Where are the results?
Regarding the reviewer comment, the table 1 was revised as provided results that include the H. pylori genome results. Results are presented in table 1.
Line 337: Section 4.4. Whole-genome sequencing and bioinformatics analysis. Where is bioinformatics analysis? Where is the additional information file?
Since the section 4.4 is material method section, bioinformatic tools used in analysis were presented with references (Table 3). However, regarding the comments of the reviewer, we added the additional information file as supplement data in results section. Ä°t is indicated in line 155-156.
The author should attach all raw data as supplementary data to support his/her results.
All raw data were attached as supplementary data.
All References need more work – there are some odd capitals/lacks of capitals.
All references were revised in terms of writing rules.
Best Regards

Reviewer 2 Report
Comments and Suggestions for Authors
Better agar dilution assays for clarithromycin on mutant 1 and 2 must be provided, because Figurte 2 is not clear at all. Mainly when the described MIC was 182 and >256 mg/L, respectively.
Several other needed changes are indicated in the enclosed file

Comments on the Quality of English LanguageSeveral paragraph are repeted and some sentences need to be revised, as indicated in the enclosed file
Author Response
Dear Sir/Madam,
First of all, thank you for your valuable insights and contributions. Following your suggestions, the English revisions of our article have been reviewed and restructured by native English-speaking professionals. The edits you indicated have been implemented as specified below.
Better agar dilution assays for clarithromycin on mutant 1 and 2 must be provided, because Figure 2 is
not clear at all. Mainly when the described MIC was 128 and >256 mg/L, respectively.
The figure 2 has been revised again to be explanatory and of high resolution.
Several other needed changes are indicated in the enclosed filer.
All indented changes in enclosed file were done and highlighted in the revised uploaded file.
All desired changes in enclosed file sent from reviewer were done and highlighted in the revised uploaded file.
In line 106 the paragraph revised as “
” Line 129: mg/L revised as μg/ml
Line 150: “mutation in a gene” revised for define the mutation as “single nucleotide mutation in a gene”
Line 152: the sentence “In the mutant 2 strain, mutation A2142G was detected. In addition, mutation in DNA binding RNA polymerase also detected in mutant-2” was revised as “
”.
In table 1 the mutations in Sulfite exporter TauE/SafE family protein and DNA binding RNA polymerase were specified and presented in revised table.
The tests currently used in
laboratories are designed to detect clarithromycin resistance in H. pylori by identifying only the most common mutations (A2143G, A2142G, and A2142C) associated with resistance. Therefore, we predict that the identified candidate mutations in the study could also contribute to the development of molecular diagnostic tests by highlighting new resistance
mechanisms in the target regions.
detected. In addition, a mutation was also present in DNA -binding RNA
polymerase
In Mutant-2, the A2142G was
The sentence from line 168, which was the same as the one in line 165, has been removed from the text, and the sentence in line 165 has been
rewritten as “
reported to be resistance to clarithromycin.
"
The most crucial factor leading to treatment failure was
The numbers 29 and 149, inadvertently written in lines 176 and 183, have
been removed from the text."
The all lines spacing requiring correction have been revised.
”

Round 2
Reviewer 1 Report
Comments and Suggestions for Authors
I acknowledge the following feedback provided on a research paper:
"The current form of this paper cannot be considered for publication. However, I see value in the research approach and encourage the authors to revise their manuscript.
some points have not been addressed by the author.
Figures 2 and 3 in the paper have blurry and low-resolution images. Additionally, the author mentioned that a DNA-binding RNA polymerase mutation was detected via bioinformatics analysis of whole-genome sequencing results in the Mutant-2 strain. However, the author did not present the WGS in their main manuscript. Also, the author mentioned that during the bioinformatics analysis of the sequencing results, some rRNA 245 strand breaks were detected, but the results were not presented.
To improve the paper, the author needs to undergo professional language editing before it can be published. Further, all supplementary data are missing a legend, and the author needs to provide more information to explain each figure and table. Lastly, all references need to have correct capital letters and italic names. The author should check the relevant section in the instructions for authors for more details."
Comments on the Quality of English LanguageThe paper should undergo professional language editing before it can be published.
Author Response
Dear Reviewer,
Thank you very much for your valuable insights and contributions. The article has been revised in accordance with your suggestions and comments. I respectfully present the detailed changes in the attached file for your information.
Figures 2 and 3 in the paper have blurry and low-resolution images.
In accordance with your suggestions, the resolutions of Figure 2 and Figure 3 have been increased to 300 dpi resolution, and new versions have been uploaded.
Additionally, the author mentioned that a DNA-binding RNA polymerase mutation was detected via bioinformatics analysis of whole-genome sequencing results in the Mutant-2 strain. However, the author did not present the WGS in their main manuscript.
In accordance with your suggestions, the inadvertently omitted WGS results have been elaborated upon in detail in both Table 1 and the relevant paragraph in the discussion section of the main text. The updates have been highlighted in yellow in the revised document.
Also, the author mentioned that during the bioinformatics analysis of the sequencing results, some rRNA 245 strand breaks were detected, but the results were not presented.
Breaks in 23S rRNA have been associated with potential methylation, which has been discussed in the article. The occurrence of 23S rRNA breaks manifested as signal drops during reading in Sanger sequencing graphs. Regarding your comments, data not provided in the previous article has been presented in supplementary data in the revised manuscript. These graphs have been included as supplementary data in the newly uploaded revised manuscript.
To improve the paper, the author needs to undergo professional language editing before it can be published.
The manuscript has been edited for English language, grammar, punctuation, and spelling by Enago, the editing brand of Crimson Interactive Inc. under Double Check Editing.
The revised manuscript has been checked based on all the points highlighted by you by a senior editor who specializes in editing manuscripts under our subject area. Accordingly, all the language and grammatical errors were rectified, in addition to making changes for improved clarity, readability, word choice, sentence structure, removal of redundancies, consistency, flow, and academic style. The certificate of the editing is presented as belove.
Further, all supplementary data are missing a legend, and the author needs to provide more information to explain each figure and table.
In line with your comments, it has been noticed that table and figure descriptions were inadvertently missing in the supplementary data of the manuscript, and necessary adjustments have been made. In the updated manuscript, detailed descriptions for all images and tables found in the supplementary data file have been provided, and the supplementary data has been reorganized. Changes have been marked in yellow in the respective file.
Lastly, all references need to have correct capital letters and italic names. The author should check the relevant section in the instructions for authors for more details.
Regarding your comments, references that were previously written erroneously have been reviewed again and reorganized in accordance with the format provided in the journal's author guidelines. The changes have been highlighted in yellow in the references section of the updated manuscript.

Round 3
Reviewer 1 Report
Comments and Suggestions for Authors
The authors have addressed all of my questions and the article has significantly improved.